# SOInter: A Novel Deep Energy Based Interpretation Method for Explaining Structured Output Models

**S. Fatemeh Seyyedsalehi, Mahdieh Soleymani & Hamid R. Rabiee**
Department of Computer Engineering
Sharif University of Technology
Tehran, Iran
`{fateme.ssalehi}@gmail.com`
`{soleymani,rabiee}@sharif.edu`

## Abstract

This paper proposes a novel interpretation technique to explain the behavior of structured output models, which simultaneously learn mappings between an input vector and a set of output variables. As a result of the complex relationships between the computational path of output variables in structured models, a feature may impact an output value via other output variables. We focus on one of the outputs as the target and try to find the most important features adopted by the structured model to decide on the target in each locality of the input space. We consider an arbitrary structured output model available as a black-box and argue that considering correlations among output variables can improve explanation quality. The goal is to train a function as an interpreter for the target output variable over the input space. We introduce an energy-based training process for the interpreter function, which effectively considers the structural information incorporated into the model to be explained. The proposed method's effectiveness is confirmed using various simulated and real data sets.

## 1 Introduction

The impressive prediction performance of novel machine learning methods has motivated researchers of different fields to apply these models to challenging problems. However, their complex and non-linear essence limits the ability to explain what they have learned. Interpretation gets more attention when we want to discover the reasons behind the model's decision and be sure about the trustworthiness and fairness of a trained machine learning model in areas such as medicine, finance, and judgment. Additionally, interpreting a model with a satisfying prediction accuracy in a scientific problem, which results in understanding the relationships behind the data, leads to gain new knowledge about the problem domain (Murdoch et al., 2019).

In many real-world applications, the goal is to map an input variable to a high-dimensional structured output, e.g. image segmentation, multi-label classification and object detection. In such problems, the output space includes a set of statistically related random variables. Many structured output models have been introduced which uses dependencies between output variables to increase the prediction accuracy. Many methods use graphical models, e.g. random fields, to capture the structural relations between variables. They define an energy function over these random fields, with a global minimum at the ground truth. Therefore, an inference is needed to find the best configuration of output variables regarding to the input by minimizing the energy function in the prediction step. Early efforts to utilize deep neural networks in structured output problems adopt them to extract high-level features from the input vector to calculate the energy function (Peng et al., 2009; Chen et al., 2015; Schwing & Urtasun, 2015). The computational complexity of the inference step in models that use random fields limits their ability to incorporate complex structures and interactions between output variables. Recent works in Belanger & McCallum (2016); Gygli et al. (2017); Belanger et al. (2017); Graber et al. (2018) propose to adopt deep neural networks instead of random fields to model the structure of the output space. Nevertheless, complex interactions between problem variables in such models make

their interpretation very challenging, specifically when we focus on the model behavior in predicting a single output variable.

This paper attempts to interpret a structured output model by focusing on each output variable separately. Our approach to model interpretation is based on instance-wise feature selection. Its goal is to find the subset of most important features in predicting a target output for each sample. This subset can vary across the input space. The complicated interactions between computational paths of output variables in structured output models cause critical challenges for finding a subset of important features associated with each output variable. A feature may not be used directly in the computational path of the target output but affects its value through relations with other ones. To compute the importance of a feature for a target output variable, we should aggregate its effect on all output variables correlated to this target.

Existing approaches of model interpretation can be divided into two groups, model-based and post hoc analysis (Murdoch et al., 2019). The model-based interpretation approach encourages machine learning methods that readily provide insight into the model's learning. However, it usually leads to simple models that are not sufficiently effective for complex structured output problems. Here, we follow the post hoc analysis and try to explain the behavior of a trained, structured output model provided as a black-box. Many interpretation techniques have been introduced to find the importance of features as a post hoc analysis. Works in Zhou & Troyanskaya (2015); Zeiler & Fergus (2013); Zintgraf et al. (2017) make perturbations to some features and observe their impact on the final prediction. Since we should perform a forward propagation for all possible perturbations, these techniques are computationally inefficient when we search for the most valuable features. In another trend, works in Simonyan et al. (2013); Bach et al. (2015) back-propagate an importance signal from the target output through the network to calculate the critical signal of features by calculating the gradient of the target w.r.t the input features. These models are computationally more efficient than perturbation-based techniques because they need only one pass of propagating. However, they need the structure of the network to be known. As this approach may cause a saturation problem, DeepLIFT (Shrikumar et al., 2017) proposes to propagate the difference of the output from a reference value instead of the gradient signal. In addition to these approaches, other ideas have also been introduced in model interpretation. Authors in Ribeiro et al. (2016) introduce Lime which locally trains an explainable surrogate model to simulate the behavior of a black-box model in the vicinity of a sample. It randomly selects a set of instances of the input space around that sample, obtains the black-box prediction for them, and trains the surrogate model by this new dataset. Shapley value, a concept from the game theory, explains how to distribute an obtained payout between coalition players fairly. The work in Lundberg & Lee (2017) proposes the Kernel-Shap to approximate the shapely value for each feature of a model's input as its importance for a prediction. As an information theoretic perspective on interpretation, the work in Chen et al. (2018) proposes to find a subset of features with the highest mutual information with the output as the set of the most important features.

Existing interpretation techniques can be used to explain the behavior of a structured output model, w.r.t. a single output. However, they ignore other output variables and could not utilize the correlation between output variables. This paper attempts to incorporate the structural information between output variables of a learning model, available as a black-box, while training an interpreter. We aim to present a globally-trained local interpreter which is a function over the input space that returns the index of the most important features for making decision about a target output. Since the value of other output variables affect the value of the target, incorporating structural dependencies into the training procedure of an interpreter leads to higher performance and decreases the uncertainty about the black-box behavior. This is the first time an interpreter is designed mainly for structured output models, and dependencies between output variables are incorporated during the interpreter's training. We call our method SOInter as we propose it to train an Interpreter specifically for Structured Output models.

## 2 PROPOSED METHOD

We assume a structured output prediction model is available as a black-box and there is no information about its architecture and parameters. Structured models map an arbitrary n-dimensional feature vector $\mathbf{x} \in \mathcal{X}$ to the output $\mathbf{y}^{sb} \in \mathcal{Y}$ where $\mathbf{y}^{sb} = \left[ \mathbf{y}_1^{sb}, \mathbf{y}_2^{sb}, \ldots, \mathbf{y}_d^{sb} \right]$ includes a set of correlated variables with known and unknown complex relationships and $\mathcal{Y}$ shows a set of valid configurations.

Considering $p_{\mathbf{sb}}(\mathbf{y}|\mathbf{x})$ as the distribution by which an arbitrary structured black-box predicts the output, we have,

$$\mathbf{y}^{sb} = \arg\max_{\mathbf{y}} p_{\mathbf{sb}}(\mathbf{y}|\mathbf{x}). \tag{1}$$

Now we explain our intuition about an interpreter, which explains the behavior of a structured output model in predicting a single target output variable. We focus on the $t$th dimension of the output space, $\mathbf{y}_t^{sb}$, as the target. Our goal is to train the interpreter $\mathcal{IN}_t(\mathbf{x}; \alpha)$ which explores a subset of the most important features that affect the black-box prediction for the target output $\mathbf{y}_t^{sb}$. At different localities of the input space, this subset of features may vary. Therefore, the proposed interpreter is a function $\mathcal{IN}_t(\mathbf{x}; \alpha) : \mathcal{X} \to \{0,\ 1\}^n$ over the input space with a set of parameters $\alpha$ and returns an n-dimensional $k$-hot vector in which the value of 1 shows the indices of selected $k$ important features for target output $\mathbf{y}_t^{sb}$.

The value of the model prediction for the target output $\mathbf{y}_t^{sb}$ is mostly determined by the subset of features found by a desired interpreter. Therefore, it is hardly expected changes in the values of other features lead to change in $\mathbf{y}_t^{sb}$. According to the definition of the interpreter, the vector $\mathbf{x} \odot \mathcal{IN}_t(\mathbf{x}; \alpha)$ is a perturbed version of the input vector $\mathbf{x}$ which has equal values to $\mathbf{x}$ in indices found by the interpreter. By passing these two vectors through the black-box, we expect the $t$th element of the corresponding outputs to be equal.

Consequently, we are encouraged to compare the black-box prediction for the target element of the output when a vector from the input space and its perturbed version are given to the black-box. However, since the architecture of the black-box is unknown, a loss function that directly compares these two output values can not be used to find the optimal interpreter. Therefore, in the following subsection, we attempt to obtain a penalty according to the difference between these values for the target, to train the interpreter block.

## 2.1 THE PROPOSED LOSS FUNCTION TO TRAIN THE INTERPRETER

As shown in eq. (1), $\mathbf{y}^{sb}$ is the black-box prediction for the input vector $\mathbf{x}$. We define $\tilde{\mathbf{y}}$ as the black-box prediction for the perturbed input vector,

$$\tilde{\mathbf{y}} = \arg\max_{\mathbf{y}} p_{\mathbf{sb}}(\mathbf{y}|\mathbf{x} \odot \mathcal{IN}_t(\mathbf{x}; \alpha)). \tag{2}$$

To describe inputs and their corresponding outputs of the structured black-box, we define a random field over variables $\mathbf{x}$ and $\mathbf{y}$ with an energy function $\mathcal{E}_{sb}(\mathbf{x}, \mathbf{y}; \theta)$ with a set of parameters $\theta$. For a given vector $\mathbf{x}$, the value of the energy function $\mathcal{E}_{sb}(\mathbf{x}, \mathbf{y}; \theta)$ is decreased as the probability of $\mathbf{y}$ being the corresponding black-box prediction for $\mathbf{x}$ is increased. Therefore, for a given vector $\mathbf{x}$ the energy function $\mathcal{E}_{sb}(\mathbf{x}, \mathbf{y}; \theta)$ is minimized when $\mathbf{y}$ is the black-box prediction for $\mathbf{x}$. Thus, according to the eq. (1), we have,

$$\mathbf{y}^{sb} = \arg\min_{\mathbf{y}} \mathcal{E}_{sb}(\mathbf{x}, \mathbf{y}; \theta) \tag{3}$$

and according to the eq. (2) we have,

$$\tilde{\mathbf{y}} = \arg\min_{\mathbf{y}} \mathcal{E}_{sb}(\mathbf{x} \odot \mathcal{IN}_t(\mathbf{x}; \alpha), \mathbf{y}; \theta) \tag{4}$$

As $\mathcal{IN}_t(\mathbf{x}; \alpha)$ selects substantial elements of $\mathbf{x}$ to determine the value of the $t$th dimension of the output and these elements in $\mathbf{x}$ and $\mathbf{x} \odot \mathcal{IN}_t(\mathbf{x}; \alpha)$ are equal, it is expected that the $t$th element of $\tilde{\mathbf{y}}$ and $\mathbf{y}^{sb}$ to be equal too. We substitute the $t$th element of $\tilde{\mathbf{y}}$ with the $t$th element of $\mathbf{y}^{sb}$ to obtain the new output vector $\left[\mathbf{y}_t^{sb},\ \tilde{\mathbf{y}}_{-t}\right]$ in which $\tilde{\mathbf{y}}_{-t}$ means all elements of $\tilde{\mathbf{y}}$ except of $t$th one.

Now, we compare two energy values $\mathcal{E}_{sb}(\mathbf{x} \odot \mathcal{IN}_t(\mathbf{x}; \alpha),\ \left[\mathbf{y}_t^{sb},\ \tilde{\mathbf{y}}_{-t}\right]; \theta)$ and $\mathcal{E}_{sb}(\mathbf{x} \odot \mathcal{IN}_t(\mathbf{x}; \alpha),\ \tilde{\mathbf{y}}; \theta)$. If $\mathcal{IN}_t(\mathbf{x}; \alpha)$ correctly selects important features of $\mathbf{x}$, two output vectors $\left[\mathbf{y}_t^{sb},\ \tilde{\mathbf{y}}_{-t}\right]$ and $\tilde{\mathbf{y}}$ are equal and these energy values are equal too. Otherwise, the value of $\mathcal{E}_{sb}(\mathbf{x} \odot \mathcal{IN}_t(\mathbf{x}; \alpha),\ \left[\mathbf{y}_t^{sb},\ \tilde{\mathbf{y}}_{-t}\right]; \theta)$ is greater than $\mathcal{E}_{sb}(\mathbf{x} \odot \mathcal{IN}_t(\mathbf{x}; \alpha),\ \tilde{\mathbf{y}}; \theta)$ according to the definition of the energy function. We propose to use the difference between these energy values, i.e.,

$$\mathcal{E}_{sb}(\mathbf{x} \odot \mathcal{IN}_t(\mathbf{x}; \alpha),\ \left[\mathbf{y}_t^{sb},\ \tilde{\mathbf{y}}_{-t}\right]; \theta) - \mathcal{E}_{sb}(\mathbf{x} \odot \mathcal{IN}_t(\mathbf{x}; \alpha),\ \tilde{\mathbf{y}}; \theta) \tag{5}$$

as a penalty to learn parameters of the interpreter function $\mathcal{IN}_t(\mathbf{x}; \alpha)$.

So far, we have assumed that a perfect energy function $\mathcal{E}_{sb}$ is available. We simulate the energy function with a deep neural network. We will explain about its architecture and training process in section 2.2. However, if the energy function does not describe the inputs and outputs of the black-box correctly, the energy value $\mathcal{E}_{sb}(\mathbf{x}, \mathbf{y}^{sb}; \theta)$ may not be less than $\mathcal{E}_{sb}(\mathbf{x}, \mathbf{y}; \theta)$ for some values of $\mathbf{x}$ and $\mathbf{y}$ where $\mathbf{y}$ is not equal to the black-box prediction $\mathbf{y}^{sb}$. In this situation, the loss function in eq. (5) may be less than or equal to zero for some values of $\mathbf{x}$, incorrectly. Therefore, to avoid of the propagation of the energy block fault during training the interpreter, we suggest the following loss function to update the interpreter,

$$\max\{0, \ \mathcal{E}_{sb}(\mathbf{x} \odot \mathcal{IN}_t(\mathbf{x}; \alpha), \ \left[\mathbf{y}_t^{sb}, \ \tilde{\mathbf{y}}_{-t}\right]; \theta) - \mathcal{E}_{sb}(\mathbf{x} \odot \mathcal{IN}_t(\mathbf{x}; \alpha), \ \tilde{\mathbf{y}}; \theta)\} \tag{6}$$

We also consider a fine-tuning process to update the parameters of the energy network to improve it in such situations, as we will explain in section 2.2.

In eq. (6), the value of $\tilde{\mathbf{y}}$ depends on the current interpreter $\mathcal{IN}_t(\mathbf{x}; \alpha)$ according to eq. (2). If we iteratively calculate $\tilde{\mathbf{y}}$ by eq. (2) based on the current interpreter, then we can back-propagate a gradient signal through the loss function in eq. (6) with respect to the interpreter function $\mathcal{IN}_t(\mathbf{x}; \alpha)$. We will explain the final optimization problem for training the interpreter function after presenting some details about the energy block $\mathcal{E}_{sb}$ and interpreter block $\mathcal{IN}_t(\mathbf{x}; \alpha)$ in the following subsections.

## 2.2 THE ENERGY BLOCK

The energy block $\mathcal{E}_{sb}$ is a deep neural network that evaluates the consistency of a pair $(\mathbf{x}, \mathbf{y})$ with the structural information incorporated into the input and output spaces. For an input vector $\mathbf{x}$, the value of the energy function $\mathcal{E}_{sb}(\mathbf{x}, \mathbf{y}; \theta)$ is minimized when $\mathbf{y}$ equals to the black-box prediction for $\mathbf{x}$. Different techniques to train an energy network have been introduced recently (Belanger & McCallum, 2016; Belanger et al., 2017; Gygli et al., 2017). We use this network to extract the structural information incorporated into the black-box, during training an interpreter. To the best of our knowledge, this is the first time an energy network is used to train an interpreter. In the proposed method, we train the energy network in two steps. First, in a pre-training phase, we train the energy block $\mathcal{E}_{sb}$ separately. To do this, we generate a set of training data by sampling from the input space and obtaining the black-box prediction for them. Any technique to train an energy network can be used for pre-training phase. Here, we adopt the work in Gygli et al. (2017) to pre-train the energy networks. Details of the method is described in Appendix B.

Second, in a fine-tuning step, we improve the energy function simultaneously with training the interpreter block. In the loss function of eq. (6), the energy value $\mathcal{E}_{sb}$ should be obtained for perturbed versions of samples from the input space. As interpreter is updated, these perturbed samples are from different regions of the feature space. Therefore, after each update of the interpreter, the energy block should be updated too. For an arbitrary perturbed input vector $\mathbf{x} \odot \mathcal{IN}_t(\mathbf{x}; \alpha)$, the energy value $\mathcal{E}_{sb}(\mathbf{x} \odot \mathcal{IN}_t(\mathbf{x}; \alpha), \ \mathbf{y}; \theta)$ should be minimized only when $\mathbf{y} = \tilde{\mathbf{y}}$ according to eq. (2). Otherwise, parameters of the energy network are updated using the structured hinge loss in the fine-tuning step as follows,

$$\max\{0, \ \mathcal{E}_{sb}(\mathbf{x} \odot \mathcal{IN}_t(\mathbf{x}; \alpha), \ \tilde{\mathbf{y}}; \theta) - \mathcal{E}_{sb}(\mathbf{x} \odot \mathcal{IN}_t(\mathbf{x}; \alpha), \ \mathbf{y}\prime; \theta) + m\prime\} \tag{7}$$

where $m\prime$ is a constant margin and,

$$\tilde{\mathbf{y}} = \arg\max_{\mathbf{y}} p_{\mathbf{sb}}(\mathbf{y}|\mathbf{x} \odot \mathcal{IN}_t(\mathbf{x}; \alpha)) \tag{8}$$

$$\mathbf{y}\prime = \arg\min_{\mathbf{y}} \mathcal{E}_{sb}(\mathbf{x} \odot \mathcal{IN}_t(\mathbf{x}; \alpha), \mathbf{y}; \theta) \quad s.t.: \ \mathbf{y} \neq \tilde{\mathbf{y}} \tag{9}$$

The main advantage of training in two phases is that we can initialize the energy block in the second step with parameters obtained in the pre-training phase. It increases the probability of the convergence and success of the iterative training process. Also, it speeds up the training process; therefore, the optimal interpreter is obtained with a smaller number of iterations.

## 2.3 THE INTERPRETER BLOCK

The interpreter $\mathcal{IN}_t(\mathbf{x}; \alpha)$ includes a deep neural network $\mathcal{W}_\alpha$, with a set of parameters $\alpha$, followed by a Gumbel-Softmax (Jang et al., 2017) unit. The output of this block is a $k$-hot vector that its

size equals to the size of the input vector $\mathbf{x}$. The size of the output of the deep neural network $\mathcal{W}_\alpha$ also equals to the size of the $\mathbf{x}$ and shows the importance of the elements of the feature vector $\mathbf{x}$. We use the Gumbel-Softmax unit to encourage the interpreter to find $k$ numbers of top important features for the target output. To this end, we consider the output of $\mathcal{W}_\alpha(\mathbf{x})$ as parameters of the categorical distribution. Then, we independently draw $k$ samples from this categorical distribution. Each sample is a one-hot vector in which the element with the value of 1 shows the selected feature. To have a $k$-hot vector, we can simply get the element-wise maximum of these one-hot vectors. As this sampling process is not differentiable, the Gumbel-Softmax trick proposes a continuous approximation of it. Considering following random variables,

$$\mathbf{g}_i = -\log(-\log(\mathbf{u}_i)) \quad i = 1, \ldots, n \tag{10}$$

where $\mathbf{u}_i \sim \text{Uniform}(0, 1)$, we can use the re-parameterization trick instead of direct sampling from $\mathcal{W}_\alpha(\mathbf{x})$ as follows:

$$\mathbf{c}_i = \frac{\exp\{\log(\mathcal{W}_\alpha(\mathbf{x})_i + \mathbf{g}_i)/\tau\}}{\sum_{j=1}^n \exp\{\log(\mathcal{W}_\alpha(\mathbf{x})_j + \mathbf{g}_j)/\tau\}} \quad i = 1, \ldots, n \tag{11}$$

where $\tau$ is the temperature parameter. The vector $\mathbf{c}$ is the continuous approximation of the sampled one-hot vector. To have $k$ selected features, we draw $k$ vectors, $\{\mathbf{c}^j : j = 1, \ldots, k\}$, and calculate their element-wise maximum to obtain the interpreter block output as follows,

$$\mathcal{IN}_t(\mathbf{x}; \alpha)_i = \max_j \{\mathbf{c}_i^j : j = 1, \ldots, k\} \quad i = 1, \ldots, n \tag{12}$$

## 2.4 THE FINAL OPTIMIZATION PROBLEM TO TRAIN THE INTERPRETER

The desired interpreter can be described as the solution of the following optimization problem with an equality constraint,

$$\alpha_{opt} = \arg\min_\alpha \mathbb{E}_{p(x)}[\max\{0, \; \mathcal{E}_{sb}(\mathbf{x} \odot \mathcal{IN}_t(\mathbf{x}; \alpha), \; [\mathbf{y}_t^{sb}, \; \tilde{\mathbf{y}}_{-t}] \; ; \theta) - \mathcal{E}_{sb}(\mathbf{x} \odot \mathcal{IN}_t(\mathbf{x}; \alpha), \; \tilde{\mathbf{y}}; \theta)\}]$$

$$s.t. : \quad \tilde{\mathbf{y}} = \arg\max_\mathbf{y} p_\mathbf{sb}(\mathbf{y}|\mathbf{x} \odot \mathcal{IN}_t(\mathbf{x}; \alpha)) \tag{13}$$

We propose the following greedy iterative optimization procedure to learn parameters of the interpreter,

$$\alpha^{(k)} \leftarrow \alpha^{(k-1)} + \beta \; \nabla_\alpha \mathbb{E}_{p(x)}[\max\{0, \; \mathcal{E}_{sb}(\mathbf{x} \odot \mathcal{IN}_t(\mathbf{x}; \alpha), \; [\mathbf{y}_t^{sb}, \; \tilde{\mathbf{y}}_{-t}^{(k-1)}] \; ; \theta^{(k-1)})$$

$$-\mathcal{E}_{sb}(\mathbf{x} \odot \mathcal{IN}_t(\mathbf{x}; \alpha), \; \tilde{\mathbf{y}}^{(k-1)}; \theta^{(k-1)})\}] \tag{14}$$

$$\tilde{\mathbf{y}}^{(k)} = \arg\max_\mathbf{y} p_\mathbf{sb}(\mathbf{y}|\mathbf{x} \odot \mathcal{IN}_t(\mathbf{x}; \alpha^{(k)})) \tag{15}$$

$$\mathbf{y}\prime^{(k)} = \arg\min_\mathbf{y} \mathcal{E}_{sb}(\mathbf{x} \odot \mathcal{IN}_t(\mathbf{x}; \alpha^{(k)}), \; \mathbf{y}; \theta^{(k-1)}) \quad s.t. : \; \mathbf{y} \neq \tilde{\mathbf{y}}^{(k)} \tag{16}$$

$$\theta^{(k)} \leftarrow \theta^{(k-1)} + \beta\prime \; \nabla_\theta \mathbb{E}_{p(x)}[\max\{0, \; \mathcal{E}_{sb}(\mathbf{x} \odot \mathcal{IN}_t(\mathbf{x}; \alpha^{(k)}), \; \tilde{\mathbf{y}}^{(k)}; \theta)$$

$$-\mathcal{E}_{sb}(\mathbf{x} \odot \mathcal{IN}_t(\mathbf{x}; \alpha^{(k)}), \; \mathbf{y}\prime^{(k)}; \theta) + m\prime\}] \tag{17}$$

where $\mathbf{y^{sb}} = \arg\max_\mathbf{y} p_\mathbf{sb}(\mathbf{y}|\mathbf{x})$ and $\beta$ and $\beta\prime$ are constant parameters.

At the first step of each iteration, the parameters of the interpreter block are updated according to the proposed loss function introduced in eq. (6). Then, the black-box output for perturbed versions of the input vectors is calculated in the second step. As mentioned, the energy function should be minimized only when $\mathbf{y} = \tilde{\mathbf{y}}$. To preserve this property for the energy function, we fine-tune it in the third and fourth steps. In the third step, we search the output space except for $\tilde{\mathbf{y}}$ to find $\mathbf{y}\prime$ for which the energy function has the minimum value. Ideally, the value of energy for $\tilde{\mathbf{y}}$ should be less than the value of energy for $\mathbf{y}\prime$. Otherwise, a penalty is considered for the energy function in the fourth step, and its parameters are updated.

The initial value $\alpha^{(0)}$ is randomly selected and its associated $\tilde{\mathbf{y}}^{(0)}$ is obtained using the second step. The energy network is initialized with the pre-trained network obtained in the pre-training phase described in section 2.2. The algorithm is continued until the value of the penalty does not considerably change which is usually obtained in less than 100 iterations. Fig. 3 in Appendix A shows an overview description of the proposed method.

## 3 EXPERIMENTS

We compare the performance of SOInter with two well-known interpretation techniques, Lime (Ribeiro et al., 2016) and Kernel-Shap (Lundberg & Lee, 2017), which are frequently used to evaluate the performance of interpretation methods, and L2X (Chen et al., 2018) which proposes an information theoretic method for interpretation. None of these techniques are specifically designed for explaining structured output models. Indeed, they only consider the target element of the output and ignore other ones during interpretation. We also evaluate the performance of SOInter-1, a version of SOInter in which no structural information is incorporated during training interpreters. In SOInter-1 the input vector and only the target output are given to the energy network and the value of other outputs are ignored during training the interpreter for the target output. Experiments are performed on both synthetic and real datasets. In section 3.1, we define two arbitrary energy functions to synthesize structured data. In section 3.2, the efficiency of SOInter is shown by explaining multi-label classifiers trained on text datasets. Finally, we evaluate the performance of SOInter by explaining a model trained for the image segmentation task in section 3.3.

### 3.1 SYNTHETIC DATASET

Here, we define two arbitrary energy functions on an input vector $\mathbf{x}$ and output vector $\mathbf{y}$, i.e. $\mathcal{E}_1$ and $\mathcal{E}_2$ in eq. (18) and (19), which are linear and non-linear functions of the input features, respectively.

$$\mathcal{E}_1 = (\mathbf{x}_1\mathbf{y}_1 + \mathbf{x}_4)(1 - \mathbf{y}_2) + (\mathbf{x}_2(1 - \mathbf{y}_1) + \mathbf{x}_3)\mathbf{y}_2 \tag{18}$$

$$\mathcal{E}_2 = (\sin(\mathbf{x}_1)\mathbf{y}_1\mathbf{y}_3 + |\mathbf{x}_4|)(1 - \mathbf{y}_2)\mathbf{y}_4 + \left(\exp(\frac{\mathbf{x}_2}{10} - 1)(1 - \mathbf{y}_1)(1 - \mathbf{y}_3) + \mathbf{x}_3\right)\mathbf{y}_2(1 - \mathbf{y}_4) \tag{19}$$

$\mathcal{E}_1$ describes the energy value over a structured output of length 2 and $\mathcal{E}_2$ is for an output of length 4. Input features are randomly generated using the standard normal distribution. In each scenario, we simulate input vectors with the size of 5, 10, 15 and 20. Outputs are vectors of binary discrete variables. The corresponding output of each input vector is found by minimizing the energy function from which we attempt to generate data. For each synthetic dataset, we train a structured prediction energy network (Gygli et al., 2017) as a black-box which has the sufficient ability to learn energy functions in eq. (18) and (19). Therefore, we can assume it has successfully captured the important features with a negligible error rate.

The SOInter architecture we use in this experiment is as follows. The energy function in the energy block is calculated as follows,

$$\mathcal{E}(\mathbf{x}, \ \mathbf{y}) = \sum_{i=0}^{n} \mathbf{y}_i \mathbf{a}_i^T F(\mathbf{x}) + \mathbf{b}^T g(\mathbf{B}\mathbf{y}) \tag{20}$$

where vectors $\mathbf{a}_i$s, $\mathbf{b}$ and matrix $\mathbf{B}$ are trainable parameters and $g(.)$ is the Softplus non-linearity function. $F(\mathbf{x})$ extracts features from the input vector, including two successive 150-dimensional fully connected layers with the Softplus non-linearity function. In eq. 20, the first term evaluates the consistency between the input feature vector and each output variable separately. The second term measures the consistency between output variables using matrix $\mathbf{B}$. The expected correlation between output variables is extracted from the training data by learning matrix $\mathbf{B}$ during pre-training and fine-tuning steps of the energy block. The deep neural network $\mathcal{W}_\alpha$ in the interpreter block includes three 100-dimensional fully connected layers with ReLU non-linearity function and one fully connected layer with the size of input vector length followed by the Gumbel-Softmax unit.

According to simulated energy functions in eq. (18) and eq. (19), only first four features of input vectors affect the value of outputs. Therefore, the desired output of an interpreter should include these four features. We used interpretation techniques to find the subset of important features for each sample. Fig. 1 compares the performance of interpreters for three arbitrary target outputs in synthesized datasets. First three diagrams of fig. 1 show the accuracy obtained by each method. As the accuracy measures the exact match of the subset of important features with the ground truth, we also calculate the median rank of results (Chen et al., 2018). To this end, we consider an order for input features and report the median rank of indices of obtained important features using each method. As the first four features are the solution, the desired median rank is 2.5. SOInter has an overall better performance compared to others and has the nearest median rank to 2.5 in the most cases. Results of SOInter-1 is also acceptable compared to other methods. However, SOInter has a better

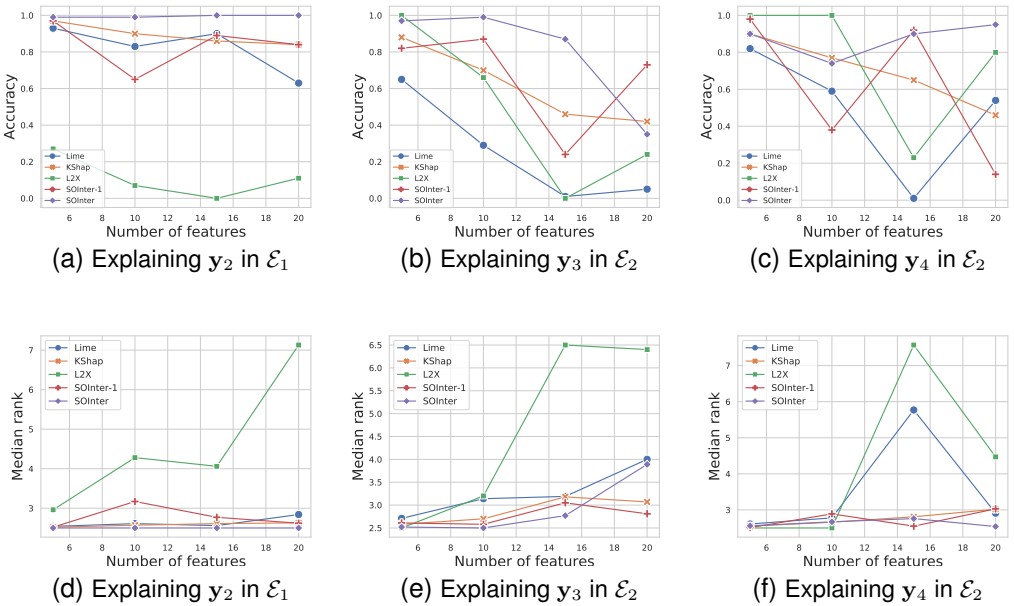

Figure 1: The accuracy ((a), (b) and (c)) and median rank ((d), (e) and (f)) obtained by Lime, Kernel-Shap, L2X, SOInter-1 and SOInter as a function of the input vector size. The SOInter performance is overall better than others.

Table 1: Performances on explaining multi-label classifiers on text dataset

| Dataset | Method | plain $F1$ | post-hoc $F1$ | relative post-hoc $F1$ |
|---------|--------|------------|---------------|------------------------|
| **Bibtex** | Lime | 0.89 | 0.54 | 0.60 |
| | L2X | 0.89 | 0.13 | 0.13 |
| | SOInter-1 | 0.89 | 0.56 | 0.64 |
| | SOInter | 0.89 | **0.60** | **0.68** |
| **Enron** | Lime | 0.43 | 0.21 | 0.35 |
| | L2X | 0.43 | 0.15 | 0.06 |
| | SOInter-1 | 0.43 | 0.21 | 0.51 |
| | SOInter | 0.43 | **0.24** | **0.57** |
| **IMDB** | Lime | 0.05 | **0.05** | 0.12 |
| | L2X | 0.05 | 0.03 | 0.04 |
| | SOInter-1 | 0.05 | **0.05** | **0.31** |
| | SOInter | 0.05 | **0.05** | 0.24 |

performance compared to SOInter-1 in most cases which confirms the importance of incorporating structural information during training an interpreter.

As the number of input features is increased, the performance of methods is generally degraded. This is because the ratio of important features compared to the size of the input vector is decreased, which can lead to confusion for the interpreter. However, experimental results confirm the robustness of SOInter when the size of the input vector is increased compared to the number of important features. Two properties of the training process of SOInter could cause this robustness. First, SOInter is trained globally through all the input space. Therefore, its parameters are adjusted by exploring all the input space. Second, all output variables are incorporated during the training of the SOInter for a specific target output. The interaction of output variables helps to find a more robust interpreter.

## 3.2 Explaining multi-label classifiers on text datasets

To evaluate the performance of SOInter, we explain black-box networks that are trained for the task of multi-label classification. We use three datasets on the text data type. Bibtex (Katakis et al., 2008), which contains bibtex entries from the BibSonomy system, presented with feature vectors of length 1836, annotated with a set of 159 tags. Enron (Read et al., 2008), which is based on a collection of email messages, presented with feature vectors of length 1001, that are categorized into 53 topics. IMDB (Read, 2010), which contains movie summaries from the Internet Movie Database, labeled with one or more genres. It maps input vectors of length 1001 to output vectors of length 28. In these datasets, each input feature vector describes a textual document by showing a numerical score for words according to the text. The output is a binary vector whose elements are tags that describe the topic of the text. The topic can be described with more than one tag. Here, we train a structured prediction energy network (Gygli et al., 2017) as a multi-label classifier for each dataset as a black-box. The architecture of the SOInter components, i.e., energy block and $\mathcal{W}_\alpha$, are the same as the SOInter in section 3.1.

To interpret black-boxes, we select an element of the output vector, which shows a subjective tag, as a target of explanation and find the top 30 important features, for each sample. To evaluate features selected by an interpreter, we only keep selected features in each input vector and replace other ones with zero. Then, we give these perturbed input vectors to the black-box and measure its performance. We calculate the average $F1$ over all tags and report in table 1. In this table, plain $F1$ shows the average $F1$ when original input vectors are given to the black-box and compares the ground-truth output with $\mathbf{y}^{sb}$. The post-hoc $F1$ shows the average $F1$ when perturbed input vectors are given to the black-box and compares the ground-truth output with $\tilde{\mathbf{y}}$. Finally, the relative post-hoc $F1$ compares $\tilde{\mathbf{y}}$ and $\mathbf{y}^{sb}$ and measures how the model can reconstruct its output when only selected features are given. As expected, the plain $F1$ is constant across all methods as it is independent of the interpreter. The higher post-hoc and relative post-hoc $F1$ show the higher quality of features selected by interpreters. As confirmed in table 1, SOInter-1 and SOInter perform better than other methods. When there is a significant correlation between the computational paths of output variables in the black-box, we expect SOInter to outperform SOInter-1. In Appendix E, we aggregate the interpretation results across the whole dataset and show top related features to some tags according to SOInter and Lime results.

## 3.3 Explaining image segmentation on Weizmann-horses dataset

Image segmentation is a structured output learning task in which the image is partitioned into semantic regions. Here, we use the Weizmann-horses dataset (Borenstein & Ullman, 2004) which consists of 328 images of left oriented horses and their masks that partition images into two regions which determine the horse's border. As the black-box, we train a deep value network (Gygli et al., 2017) for the task of image segmentation. The architecture of the black-box and training parameters are the same as in Gygli et al. (2017).

In this experiment, in the energy block of the SOInter, an input RGB image is concatenated with the corresponding mask. Then, it is given to three convolutional layers of kernel size 5 that contain 64, 128, and 128 filters, respectively. These layers are followed by a fully connected layer of size 400, a dropout layer of rate 0.25, and two fully connected layers of length 200 and 1, respectively. The deep neural network $\mathcal{W}_\alpha$ includes three convolutional layers of kernel size 5 that contain 64, 256, and 512 filters, respectively. They are followed by two fully connected layers of the size of twice the number of input pixels and a fully connected layer whose length equals the number of pixels. For all layers of two networks, we use the ReLU non-linearity function.

Fig. 2 shows the result of interpretation using SOInter for different inputs and target outputs. The target output is a pixel in the output mask, which shows the segmentation result for a specific image pixel. In fig. 2, red points show the location of the target output, and green points show the location of selected important features by SOInter. Selected features should be pixels of the input image which has the most effect on the result of segmentation for the target pixel. The first and second rows of Fig. 2 show these points in the input images and output masks, respectively. In the third row, we show the smallest rectangle, including all pixels that SOInter selected as important features for the target. As neighboring pixels of an image share the same semantics, we expect important features related to a target pixel will be located in its neighborhood. As shown, green points are placed in the vicinity

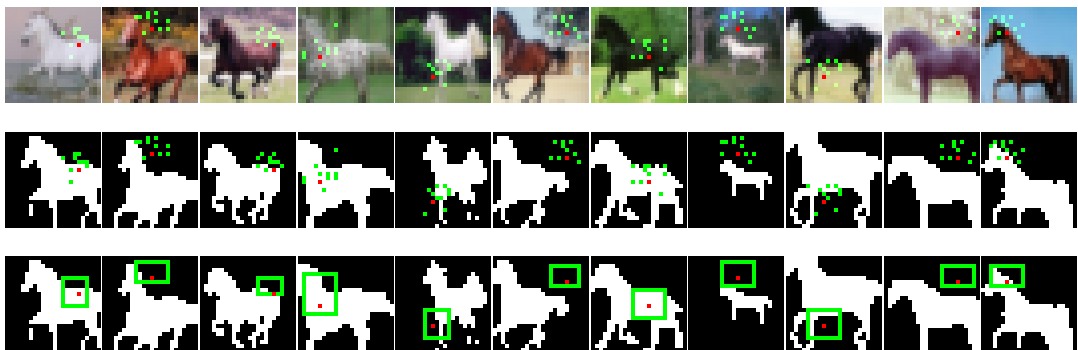

Figure 2: Samples of SOInter results on Weizmann-horses dataset. Target outputs and selected features are shown with red and green points. The first and second rows show results in the input image and its corresponding mask. The third row shows the smallest rectangle, including all selected features. Selected features are located in the locality of the target output.

of the red points, which confirms the success of SOInter. To investigate the robustness of a trained SOInter against noisy images, we corrupt input samples by different levels of noise and obtain the set of important features. Results are shown in fig. 6 of Appendix D. Generally, a trained SOInter has a satisfying robustness against noisy images.

# 4 DISCUSSION AND CONCLUSION

We have presented SOInter, an interpreter for explaining structured output models. We focus on a single element of the output of a structured model, which is available as a black-box, as a target. Then, we train a function over the input space which finds a subset of important features of the input vector which substantially affects the black-box prediction for the target output. To the best of our knowledge, this is the first time an interpreter is designed specifically for structured output models and the correlation between output variables is incorporated during explanation.

Fig. 5 in Appendix D compares the explaining time of methods as a function of input vector size. Compared to Lime and Kernel-Shap, SOInter is more efficient and has a constant explaining time across all input vector sizes which confirms the scalability of SOInter when time is important, specifically in real-world applications in which the number of features could be very large. However, as SOInter includes two neural networks, the number of its parameters is greater than the others, and it needs more memory compared to others during deployment.

To investigate the sensitivity of the performance of the SOInter to parameter $K$, we obtain the performance of the SOInter for different values of $K$ for the Weizmann-horses dataset. Fig. 7-(a) in Appendix D shows the relative post-hoc $F1$ for different values of $K$. When $K$ increases, the information to predict the target output is increased. As shown, a considerable improvement is obtained at $K$ equals to 100. After that, the performance is improved gently. It shows when $K$ equals to 100, features selected by the interpreter involve most of the information needed to predict the target. In fig 7-(b) of Appendix D, we show how the performance of the SOInter is impacted when we change the temperature parameter $\tau$ in the Gumbel-Softmax unit. In most of our experiments, the best value of $\tau$ usually equals to 10 or 100.

During training the SOInter, we perform a feature selection using the dot product with a $k$-hot vector, which leads to zero out non-important features. In many applications, zeroing unselected features is a valid choice. However, this corruption may confuse the black-box when zero values have a special meaning for the black-box. This issue is raised for all perturbation based interpretation techniques, too. We consider this fact as a possible limitation of the SOInter that can be considered for further improvement. An other interesting direction of the research is extending the SOInter to detect coarse conceptual elements as important features, which are functions of input features, instead of independently selecting a subset of features.

ACKNOWLEDGMENTS

Authors would like to thank Yasaman Ommi for her help to run some of experiments and Armin Behnamnia for his valuable comments.

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

## A    AN OVERVIEW OF THE PROPOSED METHOD

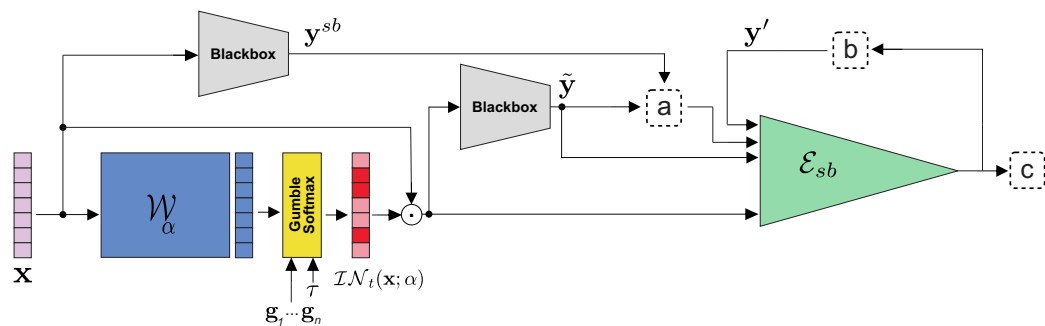

Figure 3: The overview description of the proposed method. The input vector $\mathbf{x}$ is given to the deep neural network $\mathcal{W}_\alpha$ and the Gumbel-Softmax unit, respectively. Then, the dot product of the obtained importance vector $\mathcal{IN}_t(\mathbf{x}; \alpha)$ and $\mathbf{x}$ is given to the black-box to obtain $\tilde{\mathbf{y}}$. In box (a), the value of the target output in $\tilde{\mathbf{y}}$ is substituted with $\mathbf{y}_t^{sb}$. In box (b), the minimizer of the energy function, $\mathbf{y}\prime$, is found. Finally, the calculated energy value in box (c) is used to compute the loss functions.

## B    PRE-TRAINING PHASE OF THE ENERGY NETWORK

As mentioned, any technique to train an energy network can be used for the pre-training phase. Here, we adopt the work in Gygli et al. (2017) to pre-train the energy networks. We use the following loss function to train the energy function $\mathcal{E}_{sb}$ in the pre-training phase,

$$-v\left(\mathbf{y}^{sb},\ \mathbf{y}\prime\right)\ \log\left(\sigma\left(-\mathcal{E}_{sb}\left(\mathbf{x},\ \mathbf{y}\prime; \theta\right)\right)\right) - \left(1 - v\left(\mathbf{y}^{sb},\ \mathbf{y}\prime\right)\right)\ \log\left(1 - \sigma\left(-\mathcal{E}_{sb}\left(\mathbf{x},\ \mathbf{y}\prime; \theta\right)\right)\right) \quad (21)$$

where $\sigma(.)$ shows the Sigmoid function and $\mathbf{y}^{sb}$ is the corresponding black-box prediction for sampled input vector $\mathbf{x}$ and,

$$\mathbf{y}\prime = \arg\min_{\mathbf{y}} \mathcal{E}_{sb}(\mathbf{x}, \mathbf{y}; \theta) \quad s.t.:\ \mathbf{y} \neq \mathbf{y}^{sb} \quad (22)$$

As proposed in Gygli et al. (2017), the value function $v(\mathbf{y}^{sb},\ \mathbf{y}\prime)$ measures the quality of $\mathbf{y}\prime$ compared to the expected output $\mathbf{y}^{sb}$ and could be any arbitrary function. This value function measures the distance between different output vectors and should lie between 0 and 1. *F1* and IOU metrics are valid examples of value functions for multi-label and image segmentation tasks.

## C    EXPLAINING RESULTS ON THE SYNTHETIC DATASET

In fig. 4 the explanation performance of the SOInter comparing to priors for remaining variables is reported. As in $\mathcal{E}_2$, the output variables $\mathbf{y}_1$ and $\mathbf{y}_3$ are symmetric, we ignore $\mathbf{y}_1$.

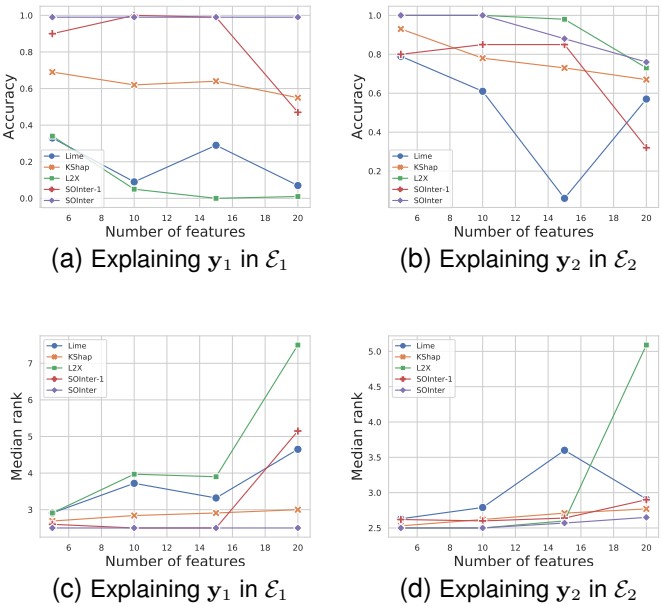

(a) Explaining $\mathbf{y}_1$ in $\mathcal{E}_1$      (b) Explaining $\mathbf{y}_2$ in $\mathcal{E}_2$

(c) Explaining $\mathbf{y}_1$ in $\mathcal{E}_1$      (d) Explaining $\mathbf{y}_2$ in $\mathcal{E}_2$

Figure 4: The accuracy ((a), (b)) and median rank ((c), (d)) obtained by Lime, Kernel-Shap, L2X, SOInter-1 and SOInter as a function of the input vector size. The SOInter performance is overall better than others.

## D  EFFICIENCY AND SENSITIVITY ANALYSIS OF THE SOINTER

Fig. 5 compares the explaining time of methods as a function of input vector size. All experiments were performed on a single NVidia-SMI-515.43.04 GPU. As shown, SOInter is more efficient than Lime and Kernel-Shap in all scenarios. Also, the explaining time of Lime and Kernel-Shap is considerably increased as the size of the input vector is increased. SOInter and L2X have a constant explaining time across all input vector sizes. It confirms the scalability of SOInter when time is important, specifically in real-world applications in which the number of features could be very large.

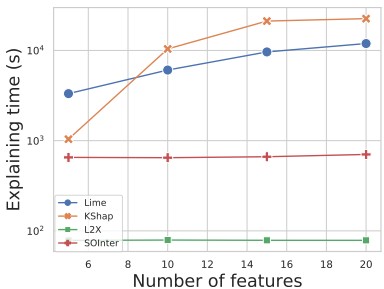

Figure 5: The time (in the log scale) of explaining 10,000 samples as a function of input vector size.

To investigate the robustness of a trained SOInter against noisy images, we corrupt images with Gaussian noises with different levels of variance. Fig. 6 shows the performance of the SOInter against noisy samples. Noise variances are shown in a logarithmic scale. As the trained SOInter have explored different rules of the black-box, it has a satisfying robustness against noise during the deployment.

To investigate the sensitivity of the performance of the SOInter on the parameter $K$, we obtain the performance of the SOInter for different values of $K$ for the Weizmann-horses dataset. Fig. 7-(a)

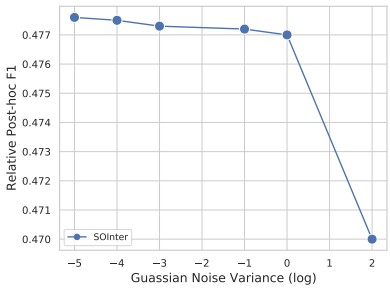

Figure 6: The sensitivity of the SOInter performance on the noise level of the input image in Weizmann-horses experiment.

shows the relative post-hoc $F1$ for different values of $K$. This measure is averaged through all pixels as the target output. We calculate these measures for features which are randomly selected, too. For small values of $K$ the information incorporated in selected features is not sufficient for the black-box to predict the output target. When $K$ increases, the information to predict the target output is increased. As shown, a considerable improvement is obtained at $K$ equals to 100. After that, the performance is improved gently. It shows when $K$ equals to 100, features selected by the interpreter involve most of the information needed to predict the target. As shown, as randomly selected features may confuse the black-box, the performance is degraded by increasing $K$ in some steps.

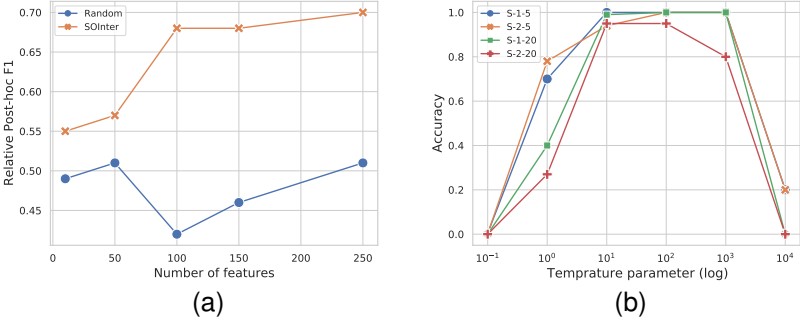

Figure 7: The sensitivity of the performance of the SOInter on the hyperparameters (a) $K$ and (b) temperature parameter $\tau$. In (a), we show the change in relative post-hoc $F1$ for the Weizmann-horses dataset when the parameter $K$ is increased. In (b) we show the change in the performance of the SOInter when $\tau$ is changed. We perform this experiment for four scenarios of synthetic datasets. S-$n$-$m$ shows results for the $\mathcal{E}_n$ and feature size of $m$.

# E  EXPLAINING MULTI-LABEL CLASSIFIERS ON TEXT DATASETS

As mentioned in section 3.2, each input vector element is associated with a word that shows its score according to a document. After interpretations, we aggregated words associated with selected important features over all samples. Then, for each word, we calculated the number of times it is chosen as an important feature for a target subjective tag across the whole dataset. Therefore, we can obtain words frequently considered an important feature by the black-box for a specific tag. In table 2, we report the 10 most frequent important words related to some tags obtained by Lime and SOInter. As shown, words found by SOInter are semantically more correlated to tags compared to those found by Lime.

Table 2: Top related features to some tags found by each interpreter

| Tag | Method | Top 10 important features related to each tag |
|---|---|---|
| **networks** | Lime | networks-network-sensor-its-decisions-free is-well-scale-modeled |
| | SOInter | topology-network-networks-nodes-learning genetic- link- scale- self |
| **education** | Lime | 24-2000 - 15 - 20 -2001- 2007-14 - 18 - 13 -2004 |
| | SOInter | educational-education-teaching-mathematics-own article-online-argue-what-school |
| **dynamics** | Lime | dynamics-2004 - 1998 - 1 - 2- 2005- networks concentration - 2001 - 06 |
| | SOInter | dynamics-simulations-molecular-phys-xxiii model-e-book-abstract-conference |
| **games** | Lime | 12-2000 - 100 - 2005 - 0- 10- 2003 - 2007 - 2002 - 13 |
| | SOInter | games-quantum-software-game-proc social-both-learning-play-often |
| **system** | Lime | as- by - an - this - that - system - we - is - this - be |
| | SOInter | system-requires-set-f-briefly-dynamic long-applications-end-operations |

