# OpenReview forum: "SOInter: A Novel Deep Energy-Based Interpretation Method for Explaining Structured Output Models"
_ICLR.cc/2024/Conference — ICLR 2024 poster_

### Official Review · Reviewer_Nt2X · 2023-10-30

**Soundness:** 2 fair
**Presentation:** 3 good
**Contribution:** 2 fair
**Rating:** 5
**Confidence:** 4

**Summary:**

In this paper, the authors propose a post-hoc interpretation method for structured output models that can only be accessed as black-boxes (ChatGPT for example, although it is not mentioned in this paper). The approach consists of two primary components: the energy block and the interpreter block, both of which are neural networks and require training. The former is an energy-based model (EBM) that is trained to evaluate the consistency of a structural input-output pair. The latter identifies the key features influencing a particular target output using a neural network followed by a Gumbel-Softmax layer. The training objective for the interpreter is to minimize the energy difference between the target output and probing output given the same subset of features. The proposed method is evaluated on both synthetic and real-world datasets, and the results show that it outperforms existing methods designed for single-output models, including LIME, SHAP, and L2X.

**Strengths:**

- Originality: The problem definition, which caters specifically to structured output models and considers structural dependencies, stands out in the field of interpretation techniques, as most existing methods are designed for single-output models. The incorporation of EBMs showcases a novel approach to model interpretation.

- Clarity & Quality: The paper is well-organized and systematically breaks down the proposed method, making it easy to follow. Definitions, equations and the role of each component are clearly explained.

- Significance: Addressing the limitations of existing interpretation techniques, especially their neglect of inter-variable dependencies in structured output models, holds significance in the broader context of model transparency and interpretability. The problem the authors tackle is particularly relevant nowadays because of the increasing popularity of LLMs.

**Weaknesses:**

- My biggest concern is that the objective function for training the interpreter seems to be valid but not well-motivated. The interpreter is trained to minimize Eq. (5), the energy difference between the target output and probing output given the same subset of features. However, it is unclear to me why minimizing this objective function makes the interpreter faithful to the structured output model. After all, the model to be explained operates on the entire input feature space, not just a subset of features. Probing the model with a small subset of features may not be a good approximation of the model's behavior because the masked inputs are almost always out-of-distribution and the model output may be arbitrary. The only connection between the interpreter and the normal operation of the structured output model is the pretraining of the energy block, which is weak. The authors should provide more justification for the proposed objective function.
- Although it is natural to use EBMs as a surrogate for structured output models, training deep EBMs is known to be difficult and less scalable than other generative models. This makes the proposed method less practical and even more unreliable.

**Questions:**

- How the constrained optimization problem in Eq. (9) & (16) is solved? Is it solved exactly or approximately? I believe this is a hard optimization problem for general EBMs.

---

> ### Author Response · Authors · 2023-11-14
>
> We thank the reviewer for their thoughtful comments on our draft. We would like to take the opportunity to focus on the highlighted weaknesses to improve the draft as follows,
>
> (1 - part 1)  The energy value measures the consistency between the value of output variables with themselves and with the input vector. As you mention, the interpreter is trained to minimize the Eq. 5 which is as follows,
> $$ \mathcal{E}\_{sb} (  \mathbf{x} \odot   \mathcal{IN}\_t(\mathbf{x};\alpha), \left[ \mathbf{y}^{sb}\_t,  \tilde{\mathbf{y}}\_{-t} \right];\theta )  - \mathcal{E}\_{sb}(\mathbf{x} \odot \mathcal{IN}\_t(\mathbf{x};\alpha),   \tilde{\mathbf{y}} ;\theta) $$
> where $\mathbf{y}^{sb}\_t$ is the $t$th element of the blackbox prediction for the raw input as follows,
> $$\mathbf{y}^{sb} = \arg\max_{\mathbf{y}} p_{\mathbf{sb}}(\mathbf{y}|\mathbf{x})$$
> and $\tilde{\mathbf{y}}\_{-t}$ are all elements except th $t$th one of the blackbox prediction for the masked input as follows,
> $$\tilde{\mathbf{y}} = \arg\max_{\mathbf{y}} p_{\mathbf{sb}}(\mathbf{y}|\mathbf{x} \odot \mathcal{IN}_t(\mathbf{x};\alpha)) $$
> In the first term of Eq. 5, the consistency of the ground truth value $\mathbf{y}^{sb}\_t$, with the sleceted features by the interpreter and also the value of the other output variables is measured. The value of other outputs, i.e. $\tilde{\mathbf{y}}\_{-t}$ are signals which include information about the blackbox.
> We select Eq. 5 as the penalty because not only selected features and the ground truth for the target output should be compatible, but also they should be consistent with the value of other output variables, which are impacted when the input is masked. It encourages the training procedure to incorporate the correlation between the output variables and also be faithful to structured models.
> By adopting the Eq. 5 as the penalty for the interpreter, the gradient which is transferred to train the interpreter, is impacted by the value of the energy function and other output variables. We will add this motivation to our new draft of the manuscript.
>
> (1 - part 2) We believe the variable $\mathbf{y}^{sb}$ in the formulation of Eq.5, which is the blackbox prediction for the raw data, can also help the training procedure to nearly overcome this issue. Also, it worth to mention that EBMs has adversarial robustness and better out-of-distribution behavior than other likelihood models [2].
>
> Generally, the issue you mentioned is also raised for all perturbation based methods. Perturbation based explanation is one of the research directions in developing interpretation techniques in which the contribution of a feature can be determined by measuring how prediction score changes when the feature is altered. Our approach can be considered a perturbation based explanation method. Solving this issue could be an interesting direction of research which recently considered in [1]. The general solution to this issue could be replacing non-selected features with a reference value (instead of simply zeroing) which does not make the data out of distribution. We will add this argument as a potential limitation of the proposed method and consider it as a further improvement in our future works.
>
> (2) We agree. However, they still have some important benefits compared to other generative models. In the case of stability, they have one module that should be trained compared to other models which need balancing between different modules. Unbalanced training
> can result in posterior collapse in VAEs or poor performance in GANs. Also, since the EBM is the only trained object, it requires fewer model parameters than approaches that use multiple networks. More importantly, the model being concentrated in a single network allows the training process to develop a shared set of features as opposed to developing them redundantly in separate networks.
> Additionally, in the proposed method, we think of energy functions as costs for a certain goals or constraints. In the case of EBMs, simply summation of energy value is a valid composition of different costs where it may not be easily practical in other generative models.
>
> Questions:
>
> (1) One of the advantage of using a deep neural network as an energy function [3] is that the inference step can be performed by iteratively backpropagation of a gradient signal with respect to the variable $\mathbf{y}$. To solve the optimization problem in eq. 6 and eq. 19, we use a simple gradient descent optimizer as follows,
> $$\mathbf{y}^{k} = \mathcal{P} ( \mathbf{y}^{k-1}  - \eta \frac{d  \mathcal{E}\_{sb} }{d\mathbf{y}} )$$
> where $ \mathcal{P} $ is the projection operator that maps the data to its feasible region and $\eta$ is set to be 0.5. As this optimization is iteratively proformed during training and pre-training the energy network, the initial value for $\mathbf{y}$ in 50% of the iterations is set to the ground truth and in 50% of iterations is randomly selected. However, this approach may lead to a local optimum.

---

> > ### Comment · Reviewer_Nt2X · 2023-11-22
> >
> > Thanks for your clarification. I would like to keep my rating unchanged.

---

> ### Author Response · Authors · 2023-11-14
>
> [1] Qiu, Luyu, et al. "Resisting out-of-distribution data problem in perturbation of xai." arXiv preprint arXiv:2107.14000 (2021)
>
> [2] Du, Yilun, and Igor Mordatch. "Implicit generation and modeling with energy based models." Advances in Neural Information Processing Systems 32 (2019).
>
> [3]Belanger, David, and Andrew McCallum. "Structured prediction energy networks." International Conference on Machine Learning. PMLR, 2016.

---

### Official Review · Reviewer_nGJs · 2023-10-31

**Soundness:** 2 fair
**Presentation:** 1 poor
**Contribution:** 2 fair
**Rating:** 3
**Confidence:** 1

**Summary:**

This paper proposes a method that focuses on the interpretation of structured outputs. The authors  use an energy-based interpretation to predict certain input that is most relevant the structured pairs. They use an greedy method to iteratively optimize the objective function and further evaluate the proposed method in several datasets.

**Strengths:**

1. interpret structured output model seems an interesting and valuable topic.

**Weaknesses:**

1. The problem under investigation appears to be of significant value. However, the author's evaluation is limited to small-scale or synthetic datasets. This raises concerns about the scalability of the proposed method and its efficacy on larger datasets.

2. Regarding the greedy approach to parameter learning, the author does not delve into an analysis of this SGD-like method nor provide relevant references. It remains unclear whether this greedy optimization truly converges to the optimal solution and how it might impact the energy model.

3. While the core idea is presented clearly, the paper's structure and flow are challenging to navigate. Additionally, the notation used lacks clarity and could benefit from further refinement.

**Questions:**

See above

---

> ### Author Response · Authors · 2023-11-22
>
> We thank the reviewer for taking time to read our work and his/her valuable comments.
>
> Weaknesses 1:
>
> In order to address your concerns regarding to the scalability of the proposed method, we would like to add the following points:
>
> 1. In summary, we attempt to evaluate SOInter for different types of data including tabular data in section 3.1, text data in section 3.2 and image data type in section 3.3. Also, we do our experiments on different values for input size and output size and also the number of important features that the interpreter should detect, i.e.
> . Additionally, we perform the explanation task through the whole datasets and report the averaged measures. Types and scales of datasets are similar to the literature [1, 2]. In section 3.2, we evaluate the performance of the SOInter on Bibtex, Enron and IMDB datasets when the number of important features is set to 30. Bibtex includes samples of feature size 1836 which maps them to the output vector of size 159. The performance of interpreters is averaged over nearly 7400 samples. Enron includes samples of features of size 1001 and map them to the output vector of size 53. For this dataset performances are averaged over 1700 samples. And finally, IMDB maps feature vector of size 1001 to output vector of size 28 and includes 120900 samples. In Section 3.3 and in the image segmentation task, images are size of 24*24 and the output space is 576. In Fig. 7-(a) of Appendix D, we show the performance of the SOInter for different values of the number of important features that the SOInter should select. Results confirm its efficiency.
>
> 2. (Regarding to the time efficiency) Indeed, one of the advantages of the SOInter is its efficiency in terms of the time complexity. For each sample, only a single forward pass through the neural network parametrizing the explainer is required to yield explanation. Thus, our algorithm is much more efficient in the explaining stage compared to other model-agnostic explainers like LIME or Kernel SHAP which require thousands of evaluations of the original model per sample. In fig. 5 of Appendix D, we show the change in the explanation time as a function of the input feature size. As shown, the SOInter explanation time doesn’t change when the number of features increases. It worths to mention that we also incorporate the training time needed for the SOInter in this figure.
>
> 3. (Regarding to the computational resources) In the training step, only the interpreter network and energy network should be loaded in the system and interacting with the black-box as an API is sufficient it shouldn’t be loaded. The size of an efficient energy network and the interpreter unit are in the same order of the black-box model. Indeed, to explore the behavior of the black-box, we do not necessarily need more neurons than the number of neurons in the black-box. In all of our experiments, the size of the energy network and the interpreter are in the same order of the size of the black-box. In the deployment step, the only module that should be loaded in the system for the interpretation task is the interpreter network.
>
> [1]Lundberg, Scott M., and Su-In Lee. "A unified approach to interpreting model predictions." Advances in neural information processing systems 30 (2017).
>
> [2]Chen, Jianbo, et al. "Learning to explain: An information-theoretic perspective on model interpretation." International conference on machine learning. PMLR, 2018.

---

> ### Author Response · Authors · 2023-11-23
>
> Weaknesses 2:
>
> Equation 13 in the manuscript describes the parameters of the optimal interpreter. However, it is not differentiable regarding to the interpreter parameters because variable $\tilde{\mathbf{y}}$ is a function of the black-box architecture that is not available ( We could only perform a prediction using the black-box and we can not do a back-propagation). Therefore, we use the subgradient method to solve this problem, in which we caculate $\tilde{\mathbf{y}}$ in equation 15 of each iteration and then caculate the subgradient of the optimization problem in equation 14. Using the subgradient approach is very common in the literature of the energy networks becaue loss functions which include energy networks are not differentiable in most cases [1].
> Structures SVM [2], structured perceptron and structures hinge loss are samples of losses on energy networks which are optimized using the subgradient method.
>
> Equation 16 and 17 are adopted from the structured hinge loss which is a very common method to train energy networks. We combine the structured hinge loss with our optimizer to revise the energy network when a conflict occcurs.
>
>
> [1] Belanger, David, and Andrew McCallum. "Structured prediction energy networks." International Conference on Machine Learning. PMLR, 2016.
>
> [2] Taskar, B., Guestrin, C., and Koller, D. Max-margin Markov networks. NIPS, 2004

---

### Official Review · Reviewer_ZVtY · 2023-11-01

**Soundness:** 3 good
**Presentation:** 3 good
**Contribution:** 2 fair
**Rating:** 6
**Confidence:** 3

**Summary:**

This work examines the problem of predicting which input features effect a specific part of a black box structured output model. The method consists of an interpreter model which outputs binary selections of input features related to a certain part of the structured output, and an energy model that approximates the structured output prediction distribution. The interpreter model sets unselected features of the input to zero. By training the interpreter to match the energy between selected features of $x$ paired with ground truth states for the relevant output structure and freely varying states for other outputs, the interpreter learns to select features of the input that are highly predictive of the relevant output structure. An algorithm for jointly learning the interpreter model and energy model is presented. Experiments on synthetic data show that model can correctly identify structured outputs when the ground truth is known, and that the method outperforms the Lime and L2X explainability methods.

**Strengths:**

* The method explores a novel angle of using an energy function to improve interpretability of structured output models.
* Experimental results show improved performance compared to the Lime and L2X methods.
* Unlike the Lime and L2X methods, the proposed method can take into account all parts of the structured model output instead of just the target element when analyzing interpretability.

**Weaknesses:**

* Even in toy examples, the model performance degrades heavily as the number of features increases, even for a relatively small amount of features such as 20.
* The Lime and L2X models used for benchmark comparison are relatively old models. It would be good to compare with more recent models if possible (although I am not an expert in this area).
* There are some practical issues setting the non-selected input states to 0, as mentioned in Section 3.3. Rather than setting states to 0, it would be better to somehow not included non-selected input states in the prediction at all. But it's not clear how to do this for certain models like ConvNets.

**Questions:**

* Are there more recent benchmarks for comparison?
* Is there a more elegant solution for suppressing non-selected input states rather than setting them to 0?

---

> ### Author Response · Authors · 2023-11-18
>
> We thank the reviewer for carefully reading the manuscript and pointing out issues may raise about the draft. We would like to take the opportunity to address mentioned weaknesses and questions as follows,
>
> Weakness 1:
>
> We agree. We provide the interpretation results of the other target outputs of the synthetic datasets in section 3.1, in the Appendix C of the new version of the manuscript, to have a better assessment about the performance of the SOInter. As shown, there is not such a considerable degradation in the performance of the SOInter compared to prior works.
> Also in section 3.2, SOInter is evaluated for larger input sizes, i.e. Bibtex with 1836, Enron and IMDB with 1001 features. Here, to have a better assessment about interpreters, the explaination performance is averaged over the whole datasets. As results confirm, SOInter has a superior performance compared to alternatives which shows its reliability for larger input sizes.
>
> In addition, we believe that in addition to the explanation accuracy, the other advantage of the SOInter is that the explanation time is not increased as size of the feature space is increased. Therefore, even in such degradation of the accuracy in some larger datasets, SOInter is still efficient in terms of the time complexity.
>
>
> Weakness 2 & Question 1
>
> Indeed, there is not. In section 3.1, we compare the proposed method with LIME, Kernel-SHAP and L2X. LIME and Kernel-SHAP are the most well-known techniques in the field of black-box model interpretation. They are frequently used for interpreting learning models in different applications and are considerably cited more than other techniques (more than 14000). Also, there are python libraries which provide a ready implementation of these algorithms to be applied easily in different applications. Other works which have been introduced recently, are back-propagation based models which need to have an access to the architecture of the model to be interpreted, and are out-of-context of our problem.
>
> In section 3.2, we compare the proposed method only with LIME and L2X. In these experiments, we average the performance of interpreters through the whole datasets. As shown in figure 2, the explaination time of Kernel-Shap is increased when the size of the input space is increased. For datasets of this section, explaining the whole datasets using Kernel-Shap takes more than 3 monthes!
>
>
>
> Weakness 3 & Question 2
>
> Generally, directly removing features from the input is impractical in practice since few models allow setting features as unknown. In addition, we assume we interpret a blackbox model that we are only able to give it a standard input and obtain its corresponding output. Therefore, we could not change the shape or the dimension of the input vector, and the only way to suppress the effect of some features is replacing their values with a reference value. For most types of data, zeroing non-important features, when the zero value has no special meaning, or replacing it with the mean value across the entire data set are both valid choices [1,2,3,4,5]. For examples, text documents are usually described by tf-idf vectors. Zeroing a features in such vectors means that the document does not include the corresponding word which is exactly equivalent to perfectly suppressing the effect of a specific feature (word).
>
> Nevertheless, occlusion raises a new concern that new evidence may be introduced and that can be used by the model as a side effect. For instance, if we occlude part of an image using green color and then we may provide undesirable evidence for the grass class. Thus, we should be particularly cautious when selecting reference values to avoid introducing extra pieces of evidence [6].
>
> [1] Chen, Jianbo, et al. "Learning to explain: An information-theoretic perspective on model interpretation." International conference on machine learning. PMLR, 2018.
>
> [2] Du, Mengnan, Ninghao Liu, and Xia Hu. "Techniques for interpretable machine learning." Communications of the ACM 63.1 (2019): 68-77.
>
> [3] Erik Štrumbelj, Igor Kononenko, and M Robnik Šikonja. Explaining instance classifications with interactions of subsets of feature values. Data & Knowledge Engineering, 68(10):886–904, 2009.
>
> [4] Luisa M Zintgraf, Taco S Cohen, Tameem Adel, and Max Welling. Visualizing deep neural network decisions: Prediction difference analysis. In International Conference on Learning Representations, 2017.
>
> [5] Piotr Dabkowski and Yarin Gal. Real time image saliency for black box classifiers. In Advances in Neural Information Processing Systems, pages 6967–6976, 2017
>
> [6] Schwab, Patrick, and Walter Karlen. "Cxplain: Causal explanations for model interpretation under uncertainty." Advances in neural information processing systems 32 (2019).

---

> > ### Comment · Reviewer_ZVtY · 2023-11-22
> > **Thanks for the discussion**
> >
> > Thanks to the authors for their valuable discussion with myself and other reviewers. I have mixed opinions about this paper. On one hand, it is a novel method is a relatively unexplored area. On the other, there are significant limitations to scaling the method to a large number of features and to apply to the method to generic architectures due to the 0 masking choice. I will raise my score to borderline accept.

---

> ### Author Response · Authors · 2023-11-22
>
> We thank the reviewer for his support of our work. In order to address your concerns regarding to the scalability of the proposed method, we would like to add the following points:
>
> 1. In summary, we attempt to evaluate SOInter for different types of data including tabular data in section 3.1, text data in section 3.2 and image data type in section 3.3. Also, we do our experiments on different values for input size and output size and also the number of important features that the interpreter should detect, i.e. $K$. Additionally, we perform the explanation task through the whole datasets and report the averaged measures. Types and scales of datasets are similar to the literature [1, 2].
> In section 3.2, we evaluate the performance of the SOInter on Bibtex, Enron and IMDB datasets when the number of important features is set to 30. Bibtex includes samples of feature size 1836 which maps them to the output vector of size 159. The performance of interpreters is averaged over nearly 7400 samples. Enron includes samples of features of size 1001 and map them to the output vector of size 53. For this dataset performances are averaged over 1700 samples. And finally, IMDB maps feature vector of size 1001 to output vector of size 28 and includes 120900 samples. In Section 3.3 and in the image segmentation task, images are size of 24*24 and the output space is 576. In Fig. 7-(a) of Appendix D, we show the performance of the SOInter for different values of the number of important features that the SOInter should select. Results confirm its efficiency.
>
>
> 2. (Regarding to the time efficiency) Indeed, one of the advantages of the SOInter is its efficiency in terms of the time complexity. For each sample, only a single forward pass through the neural network parametrizing the explainer is required to yield explanation. Thus, our algorithm is much more efficient in the explaining stage compared to other model-agnostic explainers like LIME or Kernel SHAP which require thousands of evaluations of the original model per sample. In fig. 5 of Appendix D, we show the change in the explanation time as a function of the input feature size. As shown, the SOInter explanation time doesn’t change when the number of features increases. It worths to mention that we also incorporate the training time needed for the SOInter in this figure.
>
> 3. (Regarding to the computational resources) In the training step, only the interpreter network and energy network should be loaded in the system and interacting with the black-box as an API is sufficient it shouldn’t be loaded. The size of an efficient energy network and the interpreter unit are in the same order of the black-box model. Indeed, to explore the behavior of the black-box, we do not necessarily need more neurons than the number of neurons in the black-box. In all of our experiments, the size of the energy network and the interpreter are in the same order of the size of the black-box.
> In the deployment step, the only module that should be loaded in the system for the interpretation task is the interpreter network.
>
> [1]Lundberg, Scott M., and Su-In Lee. "A unified approach to interpreting model predictions." Advances in neural information processing systems 30 (2017).
>
> [2]Chen, Jianbo, et al. "Learning to explain: An information-theoretic perspective on model interpretation." International conference on machine learning. PMLR, 2018.

---

### Official Review · Reviewer_gHPF · 2023-11-06

**Soundness:** 3 good
**Presentation:** 2 fair
**Contribution:** 3 good
**Rating:** 6
**Confidence:** 4

**Summary:**

The paper presents a novel approach to provide explainability of energy-based models in structured output prediction.  The  idea is to learn an interpreter network that predicts the k most important input variables for predicting a single output. Therefore the first contribution is to define a loss function that allows to learn such a model with the notable difficulty that the network to explain is a black-box.  The second contribution is the way the interpreter module is learned in practice since there is a non differentiability involved here. As for the architecture, the authors use a deep neural network followed by a Gumbel-Softmax unit: the re-parametrization trick allows to avoid direct sampling for the output of the neural network and replace it by a continuous approximation.
The method is showcaesd on a toy dataset and and real-world datasets in image segmentation and multilabel-classification of texts.

**Strengths:**

Strengths:
Overall, the paper is well written and reads easily.
This work presents one of the first approaches to post-hoc interpretation of structured output prediction. The proposed approach applies when the output to be predicted is a binary vector which includes a broad variety of tasks like multi-label classification, semantic segmentation... Any structured output method that predicts a bag of items for instance (bag of substructure) will be also eligible (even if not considered in this work, except for text).
The optimization problem solved to learn the interpreter is appealing with a nice way to rely on the difference of energies associated to the perturbation of inputs by the interpreter. This is really the strong novelty of the paper, for me far beyond its application to structured output prediction.

**Weaknesses:**

*The paper takes a specific angle to intepretability of structured output prediciton, by considering the input features as tabular data. When dealing with images at least, identifying "independently" the important pixels involved in the prediction is not what I expect from explainability.  I would be interesting for raw data like images by identifying a region in the image or a concept as a function of the input space as an explanation. I think a discussion here is expected.

*The learning algorithm is not sufficiently well documented and I have questions about its robustness against the choice of hyperpameter : is it robust to tau ? how does the learning algorithm react if we change k ? do we obtain close results if change k by k-1 or k+1 for instance ? What the impact of these parameters on the final "explainability"

**Questions:**

Please see questions above as well.

1) Behaviour of the learning algorithm (see previous remark).
I would like to have more insights about the behaviour of the learning algorithm - I would like to see a study about the robustness of leanring when varying k.

2)   Is it possible to incorporate in SOInter a way to encourage the identification of correlated input features for instance by taking into account the relationship between features ?
3) Did you study the robustness of your approach against noise in test images ?
4) on text multi-label classification you proposed as an evaluation metrics the post-hoc F1 and the relative F1
Please re-formulate more clearly the relative F1 (I think there is a typo).
The deceptive results may be due to the nature of  input text representation: even the words that are not considered as important by the interpreter can help to give some context and improve the performance. Can you comment on that ?

---

> ### Author Response · Authors · 2023-11-22
>
> We thank the reviewer for their thoughtful comments and suggestions. We especially thank them for their feedback on some weaknesses of the paper that can be strengthened. In this latest submission, we have addressed them as follows.
>
> Question 1 & Weaknesses 2:
>
> As per your recommendation, we perform a sensitivity analysis of the SOInter on parameter $K$ on the Weizmann-horses dataset. To this end, we consider one pixel as a target and find the top $K$ important pixels. Then, we only keep important pixels and zero pad the others and feed in the resulting masked image to the black-box to calculate the relative post-hoc F1. We report the averaged measure through all pixels as the output target. Fig. 7-(a) in Appendix D shows the obtained measures for different values of $K$. We obtain these measures for features that are randomly selected, too. For small values of $K$, the information incorporated in selected features is not sufficient for the black-box to predict the output target. When $K$ increases, the information to predict the target output is increased. As shown, a considerable improvement in the $F1$ measure is obtained at $K$ equal to 100. After that, the performance is improved gently. It shows when $K$ equals 100, features selected by the interpreter involve most of the information needed to predict the target. As shown, as randomly selected features may confuse the black-box, the performance is degraded by increasing $K$ in some steps.
>
> Finally, in Fig. 7-(b) of Appendix D, we show how the performance of the SOInter is impacted when we change the temperature parameter $\tau$ in the Gumbel-Softmax unit. In most of our experiments, the best value of $\tau$ usually equals 10 or 100.
> We add these discussions in the conclusion and discussion part of the manuscript.
>
>
> Question 2 & Weaknesses 1:
>
> Thank you for mentioning this interesting problem.  Indeed, the energy network implicitly incorporates the correlation between features while training the SOInter. The energy network measures the consistency between the value of elements of both input and output vectors. By zero padding unselected features with an interpreter, a conflict may occur in terms of the consistency between the values of features. The energy network can detect this conflict and reflect it in its value for such configuration, which impacts the training process of the SOInter.
>
> Detecting important conceptual elements behind an input vector instead of raw features is a very interesting problem that is not limited to structured output models. However, it is a difficult problem as we attempt to interpret a black-box (not white box) model. Generally, to help the SOInter find conceptual elements, we should perturb the value of neurons in the mid-layers of the energy network using the interpreter instead of masking the input features. As we also need the feedback of the black-box when these neurons are perturbed, the optimization problem becomes more complicated.  Therefore, we propose to use this version of the SOInter as a direct solution for structured output model interpretation when we need a subset of important raw features. However, as a direction for future work, we add a discussion about this interesting problem in the manuscript.
>
>
> Question 3:
>
> Thank you for this valuable comment. We add some experiments about the robustness of our trained interpreter against noisy images. In Fig. 6 of Appendix D, we evaluate the performance of the SOInter as the function of a noise variance that corrupts samples of the Weizmann-horses dataset. As the trained SOInter have explored different rules of the black-box, it has a satisfying robustness against noise during the deployment.

---

> ### Author Response · Authors · 2023-11-22
>
> Question 4:
>
> The concept of post-hoc $F1$ has been used before in [1]. In Table 1 of the manuscript, we report three $F1$ measures as follows. Considering $(\mathbf{x}, \mathbf{y}\_{gt})$ as the ground truth data from the dataset, the plain $F1$ compares $\mathbf{y}\_{gt}$s and the black-box prediction for raw inputs, i.e., $\mathbf{y}^{sb}$.
> To calculate the post-hoc $F1$, we zero pad unselected features and feed in the resulted vectors to the black-box and compares the output $\tilde{\mathbf{y}}$ with $\mathbf{y}\_{gt}$. Finally, to calculate the relative post-hoc $F1$, we zero pad unselected features and feed in the resulted vectors to the black-box and compares the output $\tilde{\mathbf{y}}$ with $\mathbf{y}^{sb}$ to evaluate the black-box ability to reconstruct its output when only selected features by the interpreter are given to the black-box. If more important features are selected, the black-box behavior is more similar to when the raw input vector $\mathbf{x}$ is given to the black-box.
>
> We agree. However, we remove unselected features before embedding $tf-idf$ vectors to obtain a text representation for a document. We zero pad unselected features in the  $tf-idf$ vectors, which perfectly suppress their effect, and then feed them to the black-box. In addition, in our evaluation, we only give the black-box selected features by interpreters. Therefore, we believe this issue is not raised in our evaluations.
>
> [1] Chen, Jianbo, et al. "Learning to explain: An information-theoretic perspective on model interpretation." International conference on machine learning. PMLR, 2018.

---

> ### Comment · Reviewer_gHPF · 2023-11-23
> **Feedback on the rebuttal**
>
> The authors provided a detailed discussion and this confirms that the paper deserves to be accepted. I'm keeping the score of 6, in comparison with other papers with even more substantial contributions.

---

### Author Response · Authors · 2023-11-22

We thank all reviewers for their helpful and valuable comments and suggestions. We applied the reviewer comments to improve the manuscript. The new changes, which are highlighted in the latest version of the uploaded draft, are as follows in summary,

1. We perform a sensitivity analysis of the optimization problem regarding to hyperparameters $K$ and $\tau$. $K$ shows the number of important features that we expect the interpreter find, and $\tau$ is the temperature parameter for the Gumbel-Softmax unit.

2. We investigate the change in the performance of the SOInter when noisy images are given to the SOInter during the deployment.

3. We add the explanation results for the SOInter and priors for other output variables of the synthetic dataset to have a better assessment about the performance of the proposed method.

4. We extend the Conclusion part of the draft to the Disscusion and Conclusion part and argue about the robustness and sensitivity of the SOInter and also argue issues suggested by reviewers.

---

### Meta-Review · Area_Chair_yUtc · 2023-12-14

**Metareview:**

The paper presents a novel approach to provide explainability of energy-based models in structured output prediction.

Strengths:
Overall, the paper is well written and reads easily. This work presents one of the first approaches to post-hoc interpretation of structured output prediction. The optimization problem solved to learn the interpreter is appealing with a nice way to rely on the difference of energies associated to the perturbation of inputs by the interpreter.

Weaknesses:
The experiments are relatively small scale.

I recommend acceptance as a poster.

**Justification For Why Not Higher Score:**

The experiments are relatively small scale.

**Justification For Why Not Lower Score:**

Overall, the paper is well written and reads easily. This work presents one of the first approaches to post-hoc interpretation of structured output prediction. The optimization problem solved to learn the interpreter is appealing with a nice way to rely on the difference of energies associated to the perturbation of inputs by the interpreter.

---

### Decision · Program_Chairs · 2024-01-16

Accept (poster)